# Parenchyma-Sparing Bronchial Sleeve Resection in Low-Grade Malignant Diseases

**DOI:** 10.3390/cancers17132156

**Published:** 2025-06-26

**Authors:** Ottavia Salimbene, Luca Voltolini, Olaf Mercier, Domenico Viggiano, Amir Hanna, Alessandro Gonfiotti, Elie Fadel

**Affiliations:** 1Division of Thoracic Surgery, Careggi University Hospital, 50134 Florence, Italy; 2Division of Thoracic and Vascular Surgery and Heart-Lung Transplantation, Marie Lannelongue Hospital, Paris-Sud University, 92350 Le Plessis Robinson, France; 3Division of Interventional Pulmunology, Department of Pulmonary Medicine, Marie Lannelongue Hospital, Paris-Sud University, 92350 Le Plessis Robinson, France

**Keywords:** sleeve bronchial resection, low-grade tumors, anastomosis

## Abstract

The study addresses a gap in current surgical approaches for treating low-grade endobronchial neoplasms. While sleeve and wedge bronchial resections that spare lung parenchyma are theoretically appealing, they are not widely adopted due to limited data. The authors conducted this research to provide evidence supporting the safety, feasibility, and effectiveness of lung-sparing bronchoplastic procedures in treating these tumors. It is a response to the need for more conservative surgical alternatives to standard lobectomy, especially for benign or low-grade malignancies.

## 1. Introduction

Since their first description by Price-Thomas [1] and Paulson and Shaw [2], sleeve resections have been increasingly adopted as safe procedures with favourable oncologic outcomes, representing a surgical option in selected cases of low-grade neoplasms, benign masses, and airway stenosis [3,4,5]. We retrospectively reviewed our experience from January 2017 to October 2024 with bronchial sleeve resections with total lung-sparing for the treatment of low-grade neoplasms. The aim of this study was to evaluate the short- and long-term outcomes of this procedure and to describe the surgical technique.

## 2. Materials and Methods

The study includes 25 bronchoplastic procedures without parenchymal resection for low-grade airway tumors performed at the Department of Thoracic Surgery of Careggi Hospital in Florence, Italy, and at the Department of Thoracic Surgery of Marie Lannelongue Hospital in Le Plessis Robinson, France, from January 2017 to October 2024.

The number of patients included in this study may appear limited, but given that we are dealing with a highly specific technique with very narrow indications, the number of reported cases is nonetheless considerable.

### 2.1. Statistical Analysis

Statistical analyses were conducted using SPSS software, version 25 (IBM Corp., Armonk, NY, USA). Continuous data were reported as medians with interquartile ranges (IQR), while categorical data were summarized as percentages. Comparisons of categorical variables were made using the Pearson χ^2^ test. For continuous variables, either Student’s *t*-test or the Mann–Whitney U test was applied, depending on data distribution. OS and DFS were estimated via the Kaplan–Meier method with 95% confidence intervals. Group differences in survival outcomes were evaluated using the log-rank test. OS was defined as the time from surgery to death from any cause, with patients censored at their most recent follow-up if death had not occurred.

DFS was defined as the period after successful treatment during which no signs or symptoms of the treated disease, nor death occurred, patients were censored at the time of the event. A *p* value ≤ 0.05 was considered significant.

### 2.2. Patient Selection

For each institution, we first conducted a search based on both the type of intervention and the definitive histological examination. Subsequently, we collected perioperative data through chart reviews and database research. Regarding follow-up, if data were missing from the database, we contacted the patients directly to update their clinical status.

We included in our series patients with a diagnosis of low-grade malignant disease, confirmed by preoperative bronchoscopy biopsy; when a biopsy was not performed, surgical planning was based on radiological criterions (computed tomography (CT) scan, 18-fluorodeoxyglucose (FDG) positron emission tomography (PET) scan and 68(Ga)-DOTATOC imaging) and on the location of the lesion. We excluded patients whose procedures included parenchymal resection and those with highly malignant tumors, as well as patients with benign conditions such as trauma, idiopathic bronchial stenosis, or complications following lung transplantation.

### 2.3. Anesthesia and Surgical Steps

As far as the surgical technique is concerned, a posterolateral serratus anterior-sparing thoracotomy was performed in all cases after a double-lumen endotracheal intubation, with the exception of a single case in which a sternotomy was performed due to a concomitant thymectomy. Once the pleural cavity was accessed, the involved bronchus was identified and carefully dissected to preserve peribronchial blood supply. The bronchus was than isolated and divided with a single sharp incision through normal tissue. This step was often guided by intraoperative bronchoscopy, to assist the surgeon in achieving an appropriate excision with adequate margins. Bronchial incisions were made perpendicular to the long axis of the airway.

Proximal and distal traction sutures were placed proximally and distally in the bronchus to reduce tension during the anastomosis. Sleeve bronchial resections were performed using a running suture technique in 12 patients, with a running suture in the membranous wall and single stitches in the cartilaginous part in 13 patients with a 3/0, 4/0 or 5/0 PDS (Polydioxanone) or Prolene. Single stitches were placed circumferentially starting from the junction between the cartilaginous and the membranous parts of the bronchus on the mediastinal side and proceeding toward the lateral side. Sutures were tied subsequentially after all had been positioned.

By precisely calibrating the spacing between sutures, it is possible to effectively compensate for even significant caliber discrepancies between the two bronchial stumps. The choice of how to calibrate the suture distance (whether using interrupted sutures or a continuous suture technique) is left to the surgeon’s discretion.

In any case, since bronchial resections and anastomoses for low-grade malignancies are typically less extensive than those for more aggressive tumors—and require smaller surgical margins—the caliber mismatch between the bronchial stumps generally does not pose a major technical challenge.

The bronchial suture was wrapped with a vital flap in 7 patients to ensure supplemental vascular support and reduce the risk of bronchovascular or bronchopleural fistula. Specifically, a pedicled pericardial fat was used in 1 case, a pleura flap in another, thymic flaps in 2 cases and the Azygos vein in 3 cases.

Once the suturing was finished, the pleural cavity was filled with saline solution, the lung was re-expanded, and a check for air leaks was conducted. Bronchoscopic evaluation verified that the suture or anastomosis was both patent and adequate in all patients.

In every case, the inferior pulmonary ligament was dissected to relieve tension on the bronchial repair. A thorough lymphadenectomy of the mediastinal and hilar nodes was consistently carried out for oncologic purposes and to facilitate the release of the structures to create a tension-free anastomosis.

A hilar lymphadenectomy combined with the removal of at least one mediastinal lymph node station was always performed, while a complete interlobar lymphadenectomy was carried out when the lesions were located outside the main left or right bronchus.

Specimens, including hilar and mediastinal lymph nodes, were sent for histologic examination. In 12 cases, frozen sections of bronchial margins were performed and in 2 cases, a further resection was carried out due of positive margins.

### 2.4. Perioperative Management

All patients were extubated immediately following surgery to minimize mechanical stress from positive-pressure ventilation on the sutures. Early ambulation and chest physiotherapy were key components of postoperative management, aimed at lowering the risk of atelectasis and pulmonary infections. Routine bronchoscopic evaluations were conducted during hospitalization, typically on the first and seventh postoperative days, to assess the integrity of the bronchial suture and rule out any signs of ischemia or stenosis.

### 2.5. Follow-Up

All patients underwent a radiologic and endoscopic follow-up according to histological results. In patients resulting R0 with no nodal involvement, a CT scan was recommended at six months and then once a year for two years followed by an annual control for ten years. For the patients resulting R1 or with nodal involvement, a multidisciplinary board decided the subsequent controls: for two patients who resulted R1 without a nodal involvement, a bronchoscopic examination was performed at three months to the operations, followed by CT scan and DOTATOC control at six months. For the patients resulted R1 and with nodal involvement, a 68(Ga)-DOTATOC and a bronchoscopic exam were performed at three months of the operation followed by CT scan, MRI and bronchoscopic exams every six months for two years and then once a year for ten years.

## 3. Results

### 3.1. Patient’s Clinical Features, Diagnosis and Preoperative Treatment

The group comprised 10 men and 15 women; the median age was 50 years, with a range of 26 to 77 years. A total of 20 patients were symptomatic, and the signs and symptoms were as follows: a total of 10 patients had cough, 8 had recurrent pneumonitis, 6 had dyspnea, 3 had chest pain, 3 had hemoptysis, and 5 patients were asymptomatic at presentation. No patient presented with carcinoid syndrome. Preoperative characteristics of the study patients are listed in Table 1.

Patient evaluation included physical examination, CT scan, and fiber optic bronchoscopy in all cases. An 18-FDG PET scan was performed in 19 cases, and a 68(Ga)- DOTATOC scan was performed in 4 patients.

Fine-needle aspiration biopsy was also performed in 21 patients. Preoperative bronchoscopic examination suggested the possibility of bronchoplastic surgery, and intraoperative bronchoscopic examination confirmed the feasibility and good quality of the bronchial suture.

Seven patients underwent bronchoscopic laser therapy via rigid bronchoscopy prior to referral, as a bridging therapy before surgery, due to tumor volume (total occlusion of a main bronchus), hemoptysis or dyspnea.

### 3.2. Surgical Indications and Type of Resection

The bronchial resection involved the left main bronchus in 14 cases, the right main bronchus in 6 cases, and the bronchus intermedius in 5 cases (Figure 1).

In 19 cases, we performed “end to end” anastomoses (Figure 2); in 5 cases, a neo-interlobar carina was created (Figure 3), and in 1 case, we performed an “end to end” anastomoses between the right main bronchus and the right lower bronchus plus a reimplantation of the right upper bronchus on the main bronchus (Figure 4).

The median operative time was 220 min, with a range to 120 min from 330 min. All the patients were extubated in the operating room.

The operative data are summarized in Table 2.

Histologic types and localizations of the resected tumors are summarized in Table 3.

Carcinoid was the most common tumor (n = 23, 92%). A biopsy was performed in 21 patients, and the specimens demonstrated 19 cases of non-specific carcinoid and 2 cases of mucoepidermoid carcinoma. In the definitive histological results, n = 20 cases were TC, n = 3 were AC, n = 2 cases were mucoepidermoid carcinoma, confirming the preoperative diagnosis.

The final histological reports showed a median length of resected bronchus of 2.1 cm (range 1.2–2.9 cm) for a median lesion dimension of 1.2 cm (range 0.3–2.5 cm). The median resection margins resulted in 0.4 cm (range 0.1–0.8 cm).

At the final histological results, three resections resulted in R1, with bronchial margins <1 mm (all of them typical carcinoids); one of them also had a nodal involvement resulting in N1 disease.

### 3.3. Postoperative Outcome

The main perioperative results and complications are summarized in Table 4.

The median length of hospital stay was 8 days (range 3–13 days), and eight patients required intensive postoperative care, with a median ICU stay of 1.6 days (range 1–5). Chest drains were removed after a median of 5 days (range 1–12 days). No intraoperative or postoperative mortality was observed.

Regarding postoperative morbidity, there were no instances of anastomosis stenosis or ischemia. One patient experienced anemia requiring two blood transfusions. Another patient required surgical intervention (thoracic lavage) due to a hemothorax. One patient developed dynamic bronchial obstruction associated with atelectasis and pneumonia, which was successfully managed with bronchoscopic toilette and antibiotic therapy.

### 3.4. Follow-Up Results

All patients underwent regular postoperative follow-up with CT scan. As of September 2024, 23 patients were still alive, while 2 had died due to causes unrelated to their bronchial neoplasm. The median follow-up duration was 37 months (range 1–88 months).

No late complications were observed; specifically, there were no bronchopleural fistula or anastomotic stenoses. During the follow-up, no patient experienced a local relapse. However, distant relapses occurred in three patients: one patient with a typical carcinoid developed a vertebral metastasis one year postoperatively, treated with radiotherapy, followed by a second vertebral carcinoid. Another patient experienced liver atypical carcinoid relapses two years after surgery, which was managed surgically. A third patient developed a mucoepidermoid cerebral metastasis one year after the surgery, treated with radiotherapy.

The five-year OS rate was 100% (Figure 5), and the five-year DFS rate was 80% (Figure 6); the graphs refer to an average follow-up of three years, as not all patients have reached the five-year follow-up.

Analysis of the DFS curve in relation to the surgical radicality revealed no local or distant recurrences among R1 patients, whereas three R0 patients experienced a distant relapse. However, this result is not statistically significant (*p* = 0.491) (Figure 7).

A further analysis was conducted by dividing patients into two groups based on resection margin length (≤0.4 or >0.4 cm): the five-year DFS appeared to favor patients with surgical margins ≤0.4 cm; however, the different was not statistically significant (*p* = 0.491) (Figure 8). Additionally, when the analysis was restricted to R0 patients only, the five-year DSF curves remained comparable between the two groups, again without statistical significance (*p* = 0.491) (Figure 9).

## 4. Discussion

Sleeve resection was first described by Price-Thomas [1] and Paulson and Shaw [2] about 60 years ago. Since then, there have been many published reports to support the use of sleeve resections [3,4,5] as a safe procedure. Although sleeve lobectomy for lung cancer was an established technique and widely used with safety, bronchoplastic procedures with total lung parenchymal sparing was an uncommon procedure. Several publications have highlighted how this procedure is adequate and safe for highly selected patients with low-grade endobronchial malignant diseases [5,6,7].

This type of technique has become an accepted surgical option for patients with benign or low-grade malignant neoplasms that do not involve the lung parenchyma, in which minimal resection margins are required to be curative [6,8,9,10], especially for tumors with a broad bronchial implantation base, full-thickness bronchial wall involvement, or tumor recurrence after endoscopic laser ablation, as well as after endoscopic complications such as bronchial stenosis or perforation [11].

Since endoscopic procedures are associated with very low morbidity and mortality, they are particularly appealing when compared to surgery [12,13]; however, the indications for endobronchial removal are very limited: only polyp-like endobronchial lesions (which represent only the 5% to 10% of the endobronchial carcinoids) without extension beyond the cartilaginous wall can be treated endoscopically [14]. Moreover, laser therapy can lead to serious complications including bleeding, tracheal or bronchial perforation and pneumothorax [15,16].

The value of operative endoscopy in the case of low-grade tumors lies in its role as a bridge to the definitive surgical treatment: aspiration and laser resection. These become essential prior to surgery to resolve atelectasis, assess the distal extent of the lesion, and, in the cases involving the carinal, to facilitate ventilation by the anesthesiologist during the surgical procedure [7,16].

In our series, six patients underwent operative endoscopy before surgery; no patient required stent placement or dilation.

Careful patient selection is crucial. The routine preoperative workup included a recent contrast-enhanced CT scan, bronchoscopy, and a physical examination [17]. The 18-FDG-PET scan was performed in 19 patients and 68(Ga)-DOTATOC in 4 patients, although the role of these imaging modalities is not well established for bronchopulmonary lesions. The 18-FDG-PET scan is useful in evaluating both typical and atypical thoracic carcinoid tumors; however, its overall sensitivity for detecting carcinoid tumors is lower than that for non-small cell lung cancer [18], with a reported sensitivity range of 14% to 100% (20). Despite the increasing evidence supporting the use of PET with somatostatin analogues [19], octreotide imaging remains relatively expensive and is not recommended for routine use in bronchopulmonary carcinoids [20]. Additionally, this specific diagnostic modality was not always available in all the medical centers.

Since bronchoscopy is mandatory in all candidates undergoing the surgical procedure [8], every patient in our study underwent bronchoscopy to define the proximal extent of the lesion and provide useful information regarding the remaining airway. Biopsy should be performed of any suspicious area, even though this procedure exposes patients to a significant risk of bleeding [17]. Nevertheless, none of our 21 patients experienced bleeding complications following the biopsy.

Regarding bronchial carcinoids, they tend to occur in patients who are, on average, about a decade younger than those with non–small cell lung cancer, and the male-to-female ratio is approximately 1:1 [21]. In our experience, the mean age was 50.9 years; it was 56.4 years in the multicenter study by Ferguson and coworkers [22], 52 years in the paper by Fink and associates [23] and 47 years in the paper by Filosso and colleagues [24]. The male-to-female ratio in our series was 0.6, and it was 1.33, 0.65, 0.57, and 0.56, respectively, in the studies by Filosso, Fink and Ferguson [22,23,24]. Most of the patients with bronchial carcinoids were symptomatic at presentation: 85% in our experience and 52% in the study by Filosso and colleagues [24].

Another fundamental step before the surgery is careful anesthesia planning. As suggested by Cerfolio et al., we preferred to use a double-lumen endotracheal tube; however, a long straight tube placed in the contralateral bronchus or a bronchial blocker positioned in the ipsilateral bronchus are alternative options [8,24].

The surgical approach was a posterolateral thoracotomy in 24 patients, as it provides access to the trachea, the mainstem bronchi, and lung [8,25]. Only 1 patient underwent a median sternotomy.

Precise attention to technical details and avoidance of extensive dissection and tension are essential for achieving excellent early and long-term outcomes. In general, the choice of anastomotic technique and suture material depends on the anatomical condition, the surgeon’s personal preference, and the experience of the surgical center.

In our study, the anastomotic techniques employed included a running suture using monofilament non-absorbable suture materials Prolene 4/0 [1,26], and a technique combining a running suture for the posterior wall with an interrupted suture for the anterior wall, using a PDS 3/0, 4/0 or 5/0, as described by Bueno et al., Cerfolio et al., and Maurizi et al. [6,9,27].

With both techniques, a size mismatch between the bronchi can be easily managed by utilizing the entire circumference in the case of the running suture and by precisely calibrating distances between stitches with the interrupted technique [1,27]. The running suture technique is simple, fast and convenient for the surgeon. However, it has recently been argued that it might lead to purse stringing [1]. The interrupted technique, on the other hand, may help reduce the risk of complete bronchial dehiscence, in cases where suture failure might occur due to infection [28,29].

Nonetheless, in an experimental study conducted by Bayram et al. [30], no significant difference was found in the rates of bronchial dehiscence or other complications in the running or interrupted technique. We support this conclusion as we observed no complications related to suture dehiscence or stenosis in our series.

Postoperative atelectasis, respiratory infection and pneumonitis remain the most common complications with an incidence of 12% [6,7]. These are primarily attributed to partial denervation of the anastomosed bronchus, ciliary epithelium damage, lymphatic obstruction, and anastomotic edema [7]. Atrial fibrillation and bleeding occur less frequently, with a reported incidence of 2–3% and 2%, respectively [6,7,9]. Air leak was reported to be 5% in the work of Bueno et al., while bronchial stenosis has been reported at a rate of 2 to 6% [31].

In our series, we observed a low overall complication rate of 16%, which included 4% respiratory infection, 4% atrial fibrillation, 4% bleeding and 4% anemia.

Bronchopleural fistula is a feared complication, with a reported incidence of 3% to 5% [6,8,9,10]. Several precautions can be taken to reduce this risk. Effective prevention includes covering the bronchial anastomosis with vital tissue and early extubation to minimize the impact of prolonged ventilatory positive pressure on the anastomosis. Various techniques have been described, including the use of mediastinal fat pad, pericardial flap, and pleural flap [32,33,34]. In our series, a protective tissue flap was used in seven cases: thymic tissue in two patients, a pleura flap in one patient, a pericardial fat in one patient, and the azygos vein in three patients.

Notably, no patients in our study developed a bronchopleural fistula, including those who did not receive flap coverage. This aligns with findings from Kutlu and Goldstraw, who, in a series of 100 patients, did not routinely wrap the bronchial anastomosis. They concluded that the meticulous airway handling and the preservation of as much peribronchial tissue as possible are sufficient to avoid the need for additional tissue coverage [35]. Similar conclusions were reported by Konstantinou et al. [36] and by Rea and colleagues [37].

Since benign and low-grade malignancies, such as those encountered in our series, require only minimal clear margins, they are ideally suited to bronchoplastic resections [5,6]. Todd et al. reviewed 67 patients with resectable carcinoid tumors and demonstrated a favorable prognosis even in cases with microscopically positive margins with no recurrence observed during 1 to 10 years of follow-up [38]. Ferguson and colleagues [22] suggested that limited resections such as wedge resections or segmentectomies should be considered for peripheral TC as local recurrence is unlikely and long-term survival is excellent.

In our series, frozen section analysis of proximal and distal margins was performed only when there was intraoperative doubt. In all other cases, resection was guided by bronchoscopic assessment alone. Final histological evaluation revealed R1 margins in only three patients, all of whom had TC tumors. In all cases, a “watch and wait” strategy was adopted, with no local adjuvant radiotherapy but close follow-up. One of them underwent a biopsy two years postoperatively due to suspected recurrence; the biopsy yielded negative results.

Indeed, there is no consensus on adjuvant therapy following resection of low-grade malignancies. Due to the relative rarity of primary bronchial carcinoids, no randomized trials have specifically addressed the benefits of adjuvant radiotherapy. As a result, clinical practice has been mainly guided by single-institution retrospective reports [39]. Schreurs and colleagues reported no recurrence following R1 bronchial carcinoid resection, with positive surgical margins showing no significant impact on prognosis, a finding consistent with previous works [40,41,42].

In addition to local tumor resection, hilar and mediastinal node dissection is mandatory for both staging and potential therapeutic benefits, especially for AC [14,40,43,44]. In our series, we routinely performed hilar and mediastinal lymphadenectomy. Only one patient, diagnosed with TC, was found to be pathologically N1. No adjuvant radiotherapy or chemotherapy was administrated and the patient remained disease-free during a two years follow-up period which is ongoing. Currently, chemotherapy is not recommended after surgical resection for patients with typical carcinoids regardless of lymph node status, due to the low risk of recurrence [45,46]. However, in patients with AC with N1 or N2 lymph node involvement, systemic recurrence is more common; in such cases, some authors advocated for adjuvant chemotherapy, although this is based on retrospective data [39,46]. Nonetheless, the optimal adjuvant regimen in this setting remains undefined.

In our experience, the five-year OS rate was 100% while recurrence occurred in three patients (12%), all of whom experienced distant relapse; this corresponds to a five-year DFS rate of 80%. These results are consistent with most reports in the literature.

Indeed, TC are associated with excellent long-term outcomes with 5- and 10-year OS rates ranging from 87% to 97% and 82% to 87%, respectively [17,23,44] and overall 5- and 10-year DFS rates of 92% and 85%, respectively [40].

In contrast, AC tumors have a significantly worse prognosis, with a 5-year survival rates ranging from 57% to 77% [17,23] with overall 5- and 10-year DFS rates of 72% and 32%, respectively [40].

For both TC and AC, distant relapses are more common compared to local relapses [25,41].

Nodal involvement has a clear impact on survival. In TC, the five-year survival rate is approximately 95% in node-negative cases and decrease to around and 75% in patients with pN1 or pN2 disease, with little difference observed between N1 and N2 involvement [11,35]. For AC, five-year survival rates are lower, about 65% without nodal involvement and 50% with pN1 or pN2 disease [17,40,47].

A previous study involving 252 patients who underwent surgery for bronchial carcinoid tumors highlighted the prognostic importance of nodal status reinforcing findings already reported in the literature [44]. In this study, Rea et al. compared two patient cohorts: Group A who underwent surgery for carcinoid tumors from 1968 to 1989 receiving a lymph node sampling and Group B receiving a systematic lymphadenectomy from 1990 to 2005. No difference was found between Group A and B in the detection of nodal metastases (10.9% vs. 11.9%, respectively), while they observed two lymph node relapses in Group A. Additionally, the rate of sleeve resections increased significantly in Group B (2.7% vs. 20.4%) and the number of pneumonectomies declined markedly (7.2% vs. 1.4%). Given the therapeutic benefit of the mediastinal lymph node dissection, even in patients with clinically positive nodes (cN1 or cN2), surgical resection with lymphadenectomy is recommended for central carcinoid tumors [14].

No cases of local relapse were observed in our series, supporting the adequacy of bronchoplastic procedure for intraluminal low-grade bronchial neoplasms. Preserving normal lung tissue remains one of our primary goals [9]. Although both local and distant recurrence are uncommon, they can occur several years after surgical resection. Therefore, long-term follow-up is essential for patients with bronchial carcinoids [36]. At present, there is no universally accepted surveillance strategy. The National Comprehensive Cancer Network guidelines recommend postoperative surveillance every 3 to 12 months initially, followed by evaluation every 6 to 12 months for up to 10 years [48]. However, such an intensive protocol may not be necessary in patients with node-negative TC, given the low recurrence risk in this population [47].

The present study has several limitations. It is a retrospective and multicenter study which includes a relatively small patient cohort and has a limited follow-up duration. Furthermore, there was no standardized treatment protocol across the participating institutions. The two hospitals involved have varying levels of expertise, and while the surgical approaches used are broadly comparable, they are not entirely uniform.

To further illustrate the clinical and decision-making process, two representative cases from our series are presented below.

### 4.1. Case 1

A 26-year-old woman with no significant comorbidities presented with recurrent right-sided pneumonia. The CT scan revealed a solid mass in the right main bronchus, at the origin of the bronchus intermedius, causing obstruction (Figure 10).

An 18-DG-PET scan revealed a small area of mild hypermetabolism (SUV max 2.5) at the site of the lesion, with no signs of metabolic hyperactivity in the mediastinal lymph node stations (Figure 11).

Based on the suspicion of a tumor, a rigid bronchoscopic examination was performed, which revealed an endobronchial mass on the right main bronchus. Histological examination of the biopsy confirmed the diagnosis of a carcinoid tumor.

Initially, laser-assisted mechanical deconstruction was performed using rigid bronchoscopy to restore patency of the right bronchial system. After a multidisciplinary team discussion, the patient underwent lung-sparing bronchial sleeve resection.

The surgery was performed via a posterolateral thoracotomy at the fifth intercostal space with left single-lung ventilation. After opening the posterior mediastinal pleura, the main, intermediate, and upper lobar bronchi were isolated using a tape (Figure 12).

During the surgery, intraoperative bronchoscopy allowed visualization of the known lesion at the level of the intermediate bronchus, near the junction of the right main bronchus and at the emergence of the right upper lobar bronchus, which helped identify the resection margin. Transection of the three bronchial segments was performed using a cold blade (Figure 13).

In order to fully mobilize the pulmonary parenchyma, the triangular ligament was divided; subsequently, the suprazygoid mediastinal pleura was opened, followed by transection of the azygos vein using an Endo GIA Tri-Staple (Covidien, Mansfield, MA, USA) with two gold 30 mm cartridges.

An end-to-side anastomosis of the intermediate bronchus at the proximal resection margin of the right main bronchus was created using a continuous suture of Prolene 4.0 (Figure 14). The right upper bronchus was then reimplanted onto the right main bronchus with an end-to-side anastomosis after creating a cold blade eyelet, also using a continuous suture of Prolene 4.0 (Figure 15). The anastomoses were covered and separated, with anchoring of the azygos stump (Figure 16).

The patient was discharged on postoperative day 8 and has been under follow-up for 1 year (still ongoing); no long-term complications occurred, and there have been no local or distant relapses.

### 4.2. Case 2

A 54-year-old female underwent a right nephrectomy in 2022 and a left mastectomy and hysteroannessectomy in 2021, followed by radiotherapy and hormonotherapy in the context of a BRCA2 mutation. She reported no fever, chest pain, loss of weight or weakness. A follow-up CT scan revealed a solid lesion with contrast enhancement in the left main bronchus (Figure 17).

There was no consolidation or volume loss of the left lung or lymphadenopathy. Functional respiratory tests were performed to evaluate airway obstruction and to assess perioperative risk with no significant findings. Additional imaging including abdominal CT and 18DG-PET did not reveal any pulmonary or extrapulmonary pathology. A 68(Ga)-DOTATOC PET showed increased uptake at the site of the level of the endobronchial mass (Figure 18).

The patient underwent rigid bronchoscopy, which revealed a rounded, well-vascularized polypoid lesion measuring 9 mm in diameter located in the left main bronchus, in proximity to the interlobar carina (Figure 19).

Histopathological examination confirmed a neuroendocrine tumor. Following multidisciplinary team discussion, the patient underwent lung-sparing bronchial sleeve resection surgery.

A muscle-sparing thoracotomy was performed through the fifth intercostal space.

After releasing the triangular ligament, the posterior and superior mediastinal pleura were opened, and the pulmonary artery trunk was dissected, controlled and secured with a multi-lumen blade. The left pulmonary artery was then separated from the bronchial plane. We dissected and controlled the superior pulmonary vein to free it from the upper lobar bronchus. The main bronchus was dissected and controlled posteriorly, while the bronchi for the upper and lower lobes were also dissected, controlled, and looped (Figure 20).

A cold blade incision was made on the main bronchus proximal to the mass (Figure 21), and the lesion was removed by performing a distal incision just before the interbronchial carina.

The specimen (Figure 22) was sent for pathological analysis; the intraoperative frozen section revealed a positive distal bronchial margin for tumor cells.

As a result, the bronchial spur was resected, releasing the bronchi of the upper and lower lobes. This time, the intraoperative frozen section confirmed that the bronchial margins was free of tumor cells. A side-to-side bronchial anastomosis was performed between the lobar bronchus using a posterior running suture with 5-0 PDS. Bronchial reimplantation into the main bronchus was then completed with a posterior running suture of 4-0 PDS, followed by interrupted anterior sutures using 4-0 and 5-0 PDS (Figure 23).

Bronchoscopic evaluation revealed good patency of the anastomosis, with all bronchi easily traversed, though a slight cartilage edge was noted near the basal pyramid. The anastomosis showed good sealing during the reventilation test.

Interlobar, hilar and mediastinal lymph node dissections were also performed, and the pathological report confirmed a pT2aN0 typical bronchial carcinoid (according to the 8th edition TNM classification) with tumor-free margins.

On postoperative day 3, bronchoscopic evaluation revealed a dynamic obstruction at the level of the anastomosis caused by a clot with purulent secretions; its removal resulted in an improvement in oxygen requirements. Four days after the operation, the patient developed fever and a worsening inflammatory response, with phlebitis being the only clinical sign. Empiric antibiotic therapy with Tazocin was initiated and later switched to Augmentin after two days, for a total duration of five days.

The chest drain was removed 3 days after the procedure following radiological confirmation. The patient was discharged on postoperative day 7 after a fibroscopic examination (Figure 24).

No short-term complications occurred.

## 5. Conclusions

In conclusion, bronchoplastic procedures without resection of the lung parenchyma are an adequate and feasible technique for selected cases of low-grade endobronchial neoplasms.

Our results are in line with those of the specific literature about this subject, in terms of the numbers of patients included and the feasibility and in terms of the short-term operative results and long-term oncological results.

Thoracic surgeons should be familiar with these techniques to provide their patients with the greatest chance to remove an obstructing lesion of the airway, preserve lung parenchyma, and provide optimal long-term functional results.

## Figures and Tables

**Figure 1 cancers-17-02156-f001:**
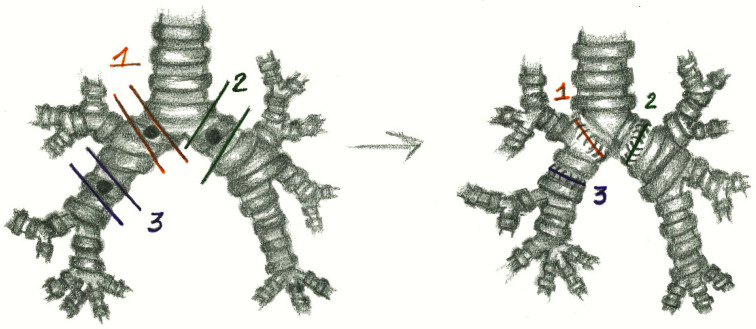
The reconstruction of the right main bronchus (1), the left main bronchus (2) and the right bronchus intermedius (3).

**Figure 2 cancers-17-02156-f002:**
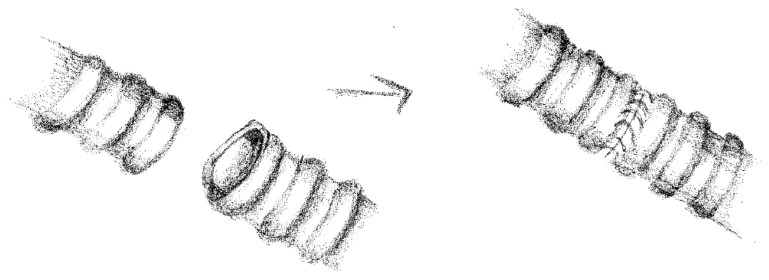
“End to end” anastomosis.

**Figure 3 cancers-17-02156-f003:**
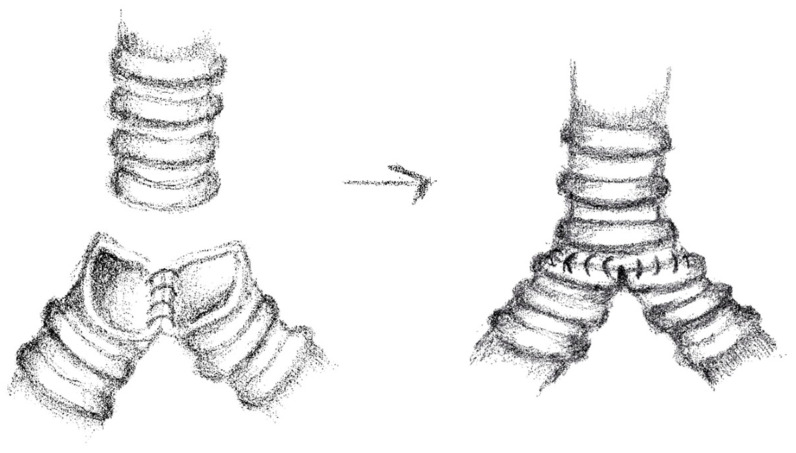
Neo-interlobar carina.

**Figure 4 cancers-17-02156-f004:**
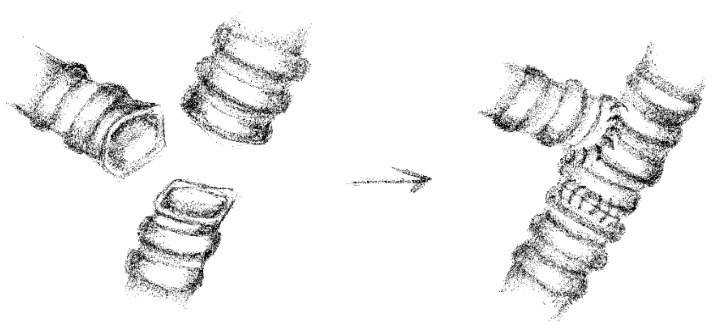
“End to end” anastomoses plus an “end-to-side” anastomosis.

**Figure 5 cancers-17-02156-f005:**
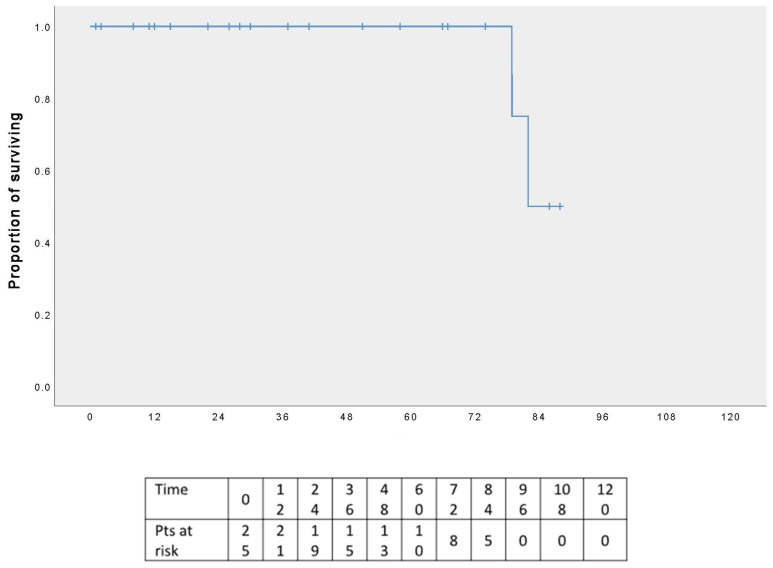
Overall survival.

**Figure 6 cancers-17-02156-f006:**
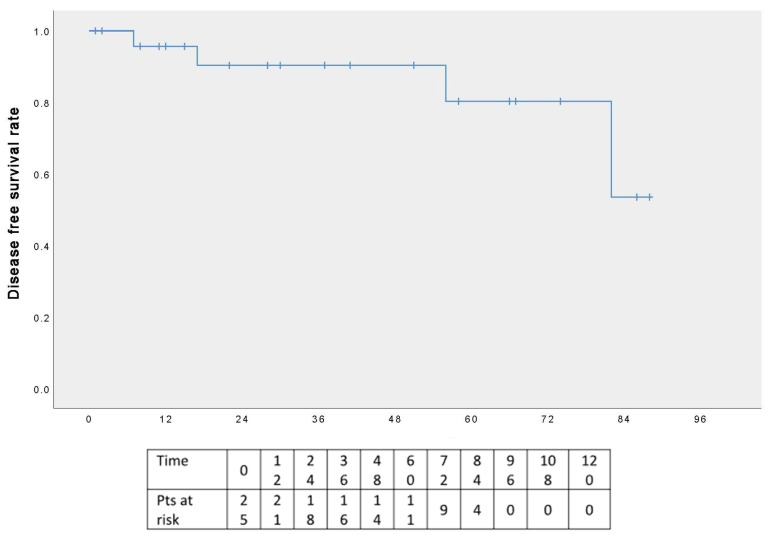
Disease-free survival.

**Figure 7 cancers-17-02156-f007:**
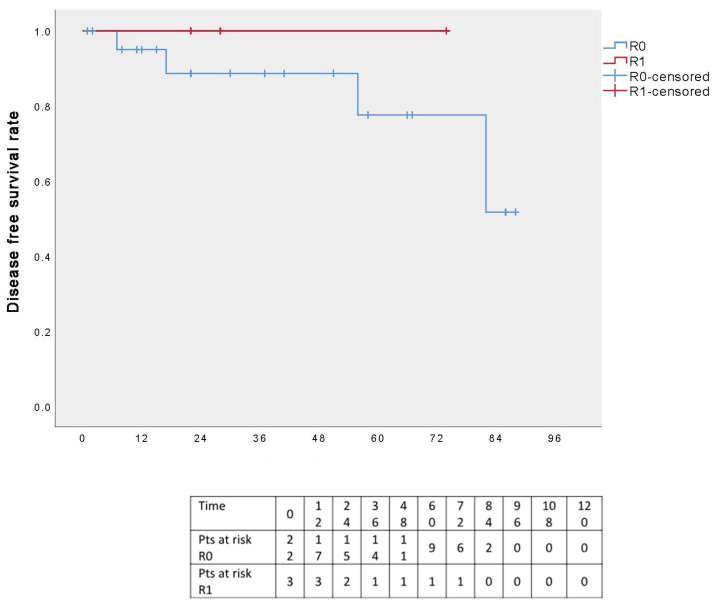
Disease-free survival in relation to surgical margins.

**Figure 8 cancers-17-02156-f008:**
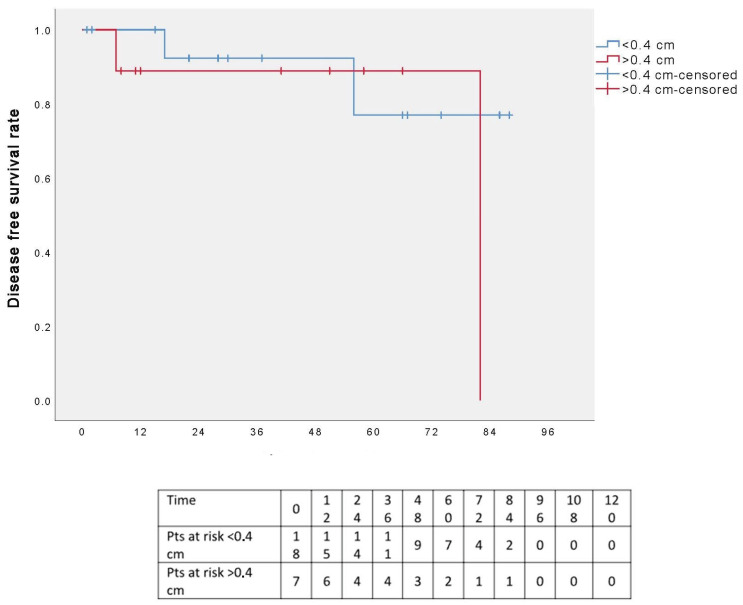
Disease-free survival in two groups according to the resection margins.

**Figure 9 cancers-17-02156-f009:**
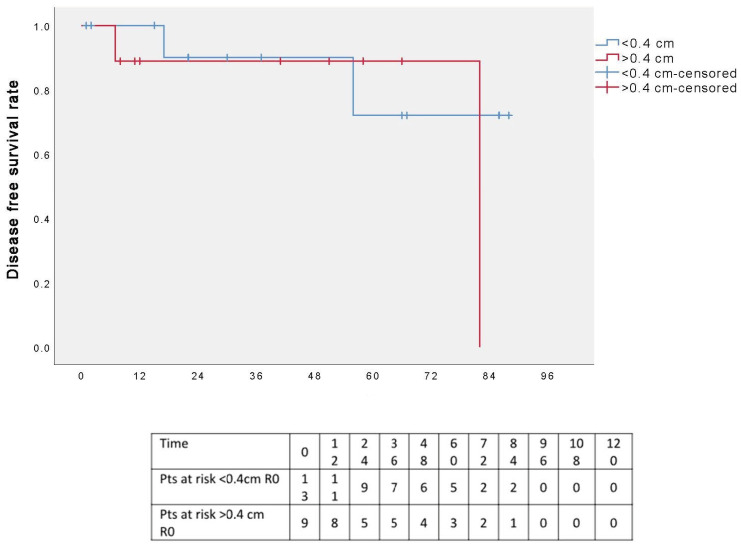
Disease-free survival in two groups according to the resection margins, selecting only R0 patients.

**Figure 10 cancers-17-02156-f010:**
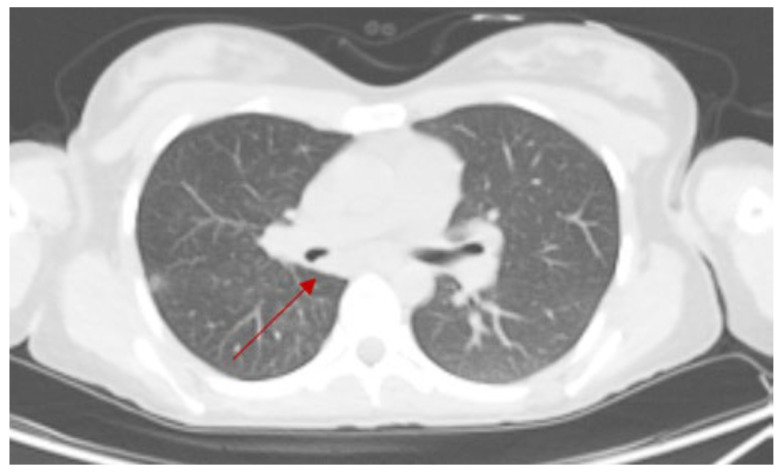
Preoperative CT scan, with a red arrow indicating the level of the mass obstructing the right main bronchus.

**Figure 11 cancers-17-02156-f011:**
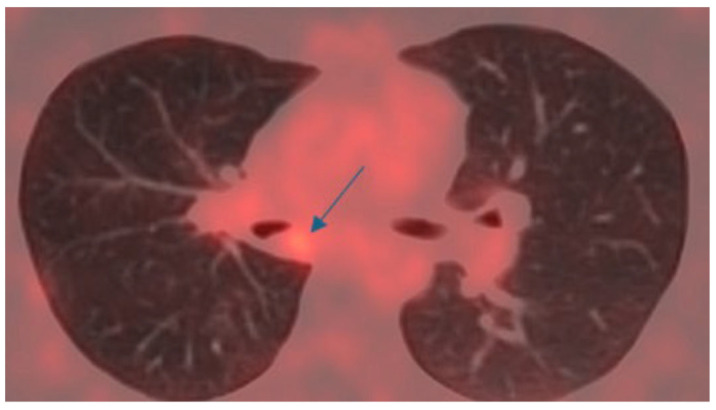
Preoperative 18-FDG-PET scan, with a blue arrow indicating the mass obstructing the right main bronchus.

**Figure 12 cancers-17-02156-f012:**
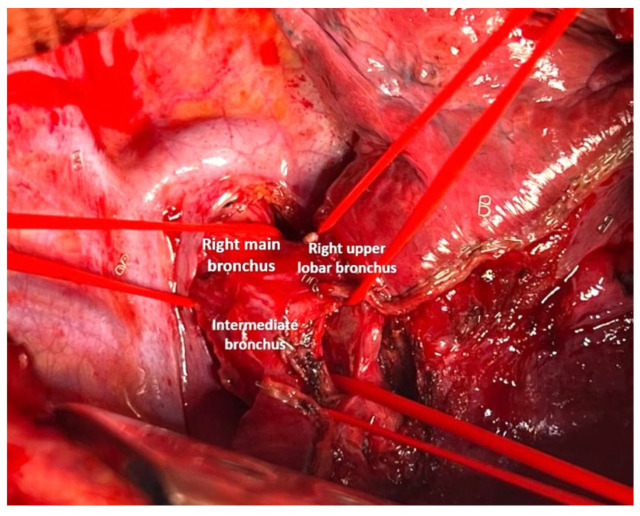
Vessel loop on the right main bronchus, the right upper lobar bronchus and the intermediate bronchus.

**Figure 13 cancers-17-02156-f013:**
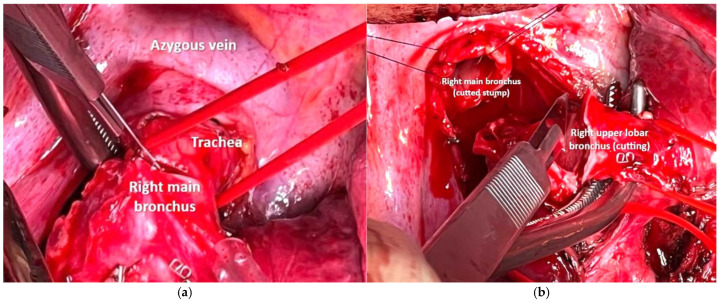
Transection with a cold blade of the right main bronchus (**a**), for the right upper lobar bronchus (**b**), for the intermediate bronchus.

**Figure 14 cancers-17-02156-f014:**
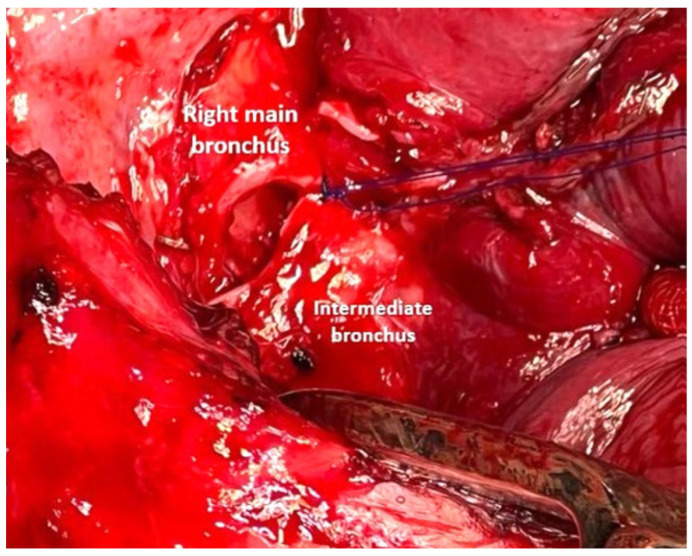
“End-to-end” anastomosis of the intermediate bronchus at the proximal resection margin of the right main bronchus.

**Figure 15 cancers-17-02156-f015:**
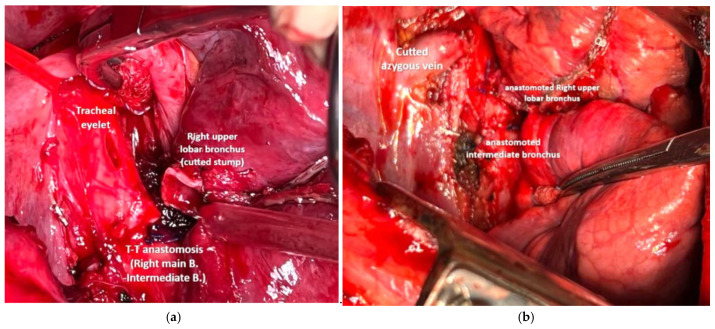
Creation of a tracheal cold blade eyelet (**a**) and the result of the final anastomosis (**b**).

**Figure 16 cancers-17-02156-f016:**
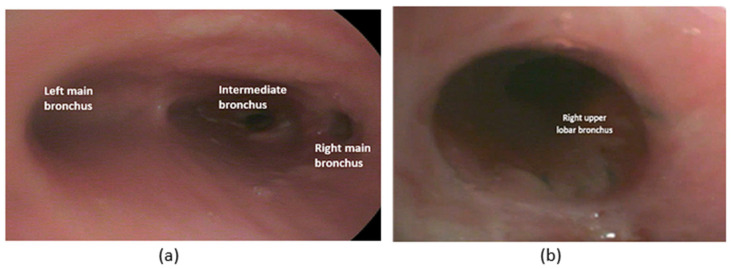
Bronchoscopic evaluation on postoperative day 7: tracheal carina (**a**), end-to-side anastomosis between the right upper bronchus and the right main bronchus (**b**), end-to-end anastomosis between the right main bronchus and the interlobar bronchus (**c**), low right bronchus (**d**).

**Figure 17 cancers-17-02156-f017:**
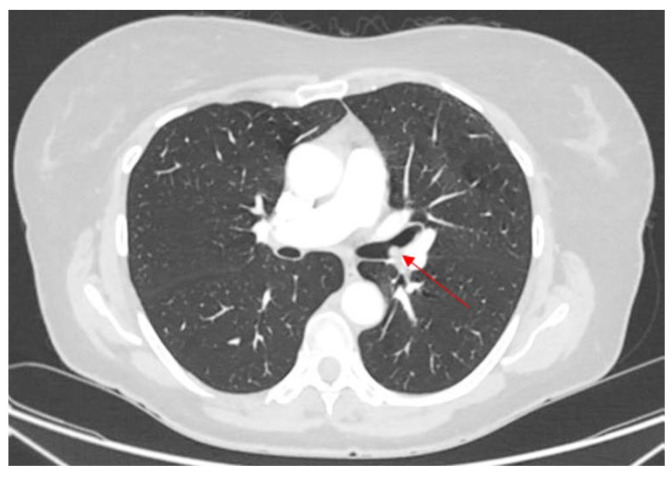
Interlobar carina endobronchial mass (red arrow).

**Figure 18 cancers-17-02156-f018:**
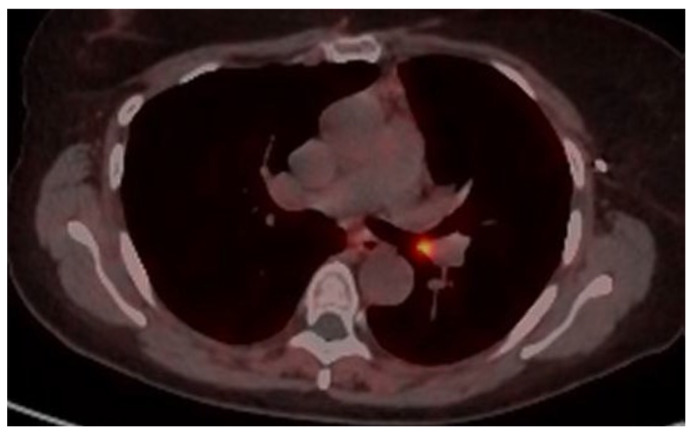
68(Ga)-DOTATOC PET scintigraphy hypermetabolism of the left bronchus lesion.

**Figure 19 cancers-17-02156-f019:**
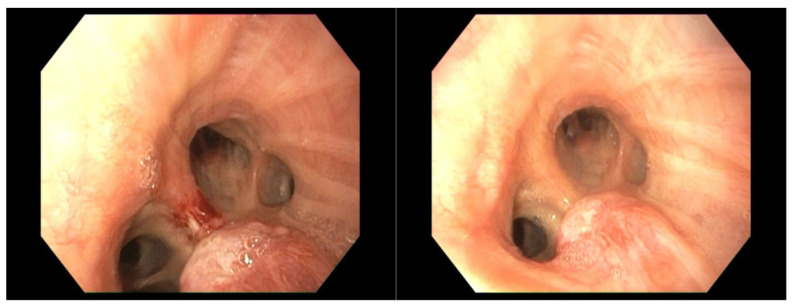
Initial bronchoscopy with an exophytic well-vascularized polypoid lesion.

**Figure 20 cancers-17-02156-f020:**
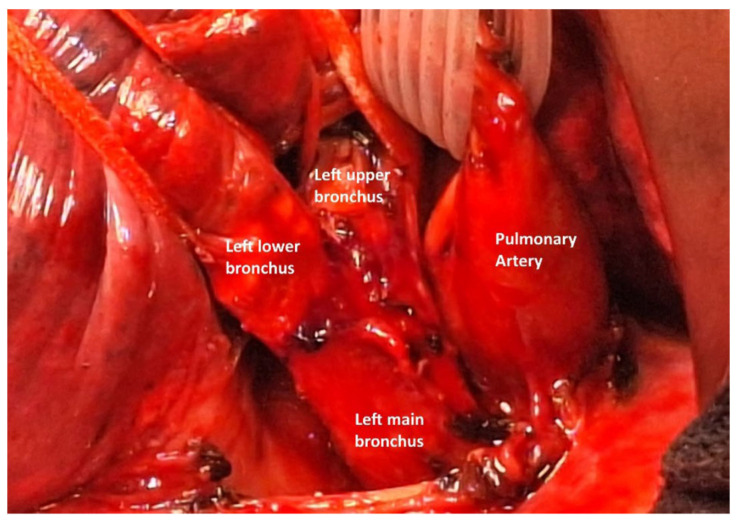
Vessel loop on the left lower bronchus, and isolation of the left upper bronchus and of the pulmonary artery.

**Figure 21 cancers-17-02156-f021:**
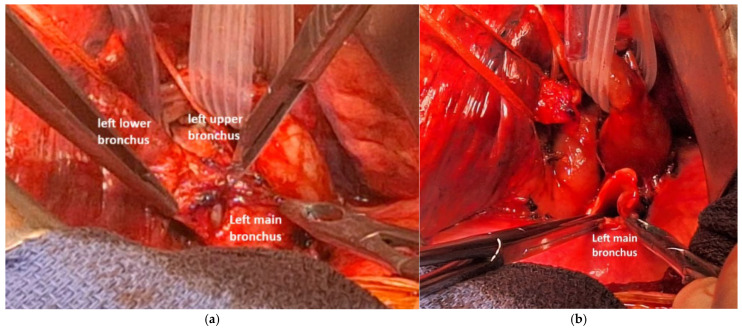
Transection with a cold blade of the left main bronchus (**a**,**b**).

**Figure 22 cancers-17-02156-f022:**
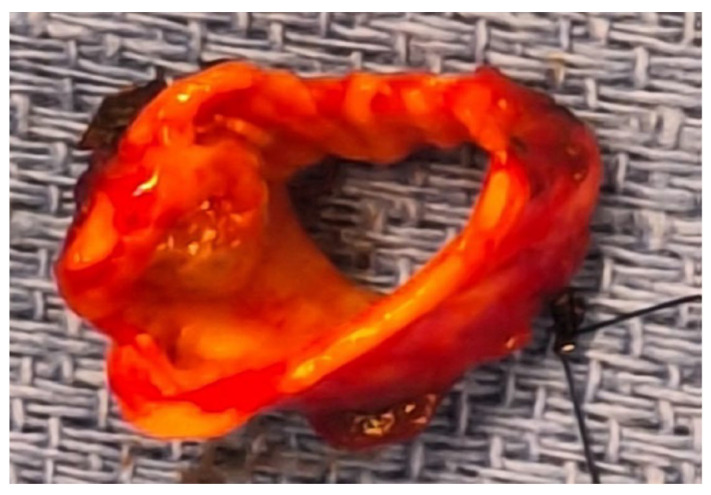
Specimen with the removed lesion.

**Figure 23 cancers-17-02156-f023:**
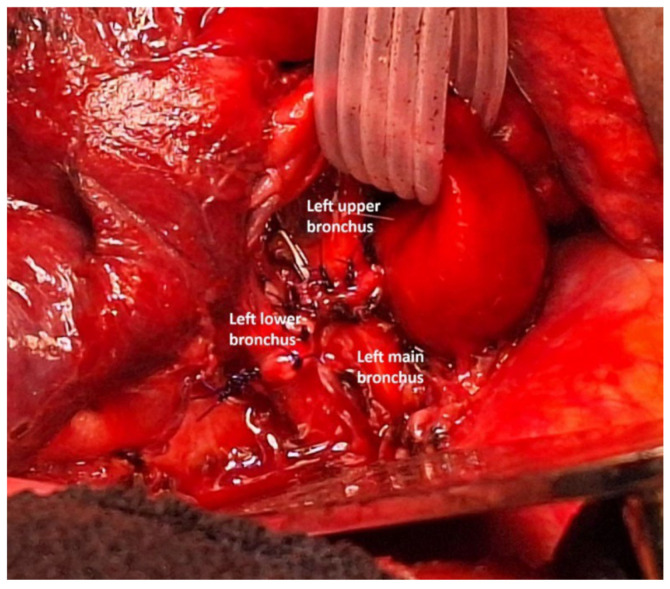
Result of the final anastomosis.

**Figure 24 cancers-17-02156-f024:**
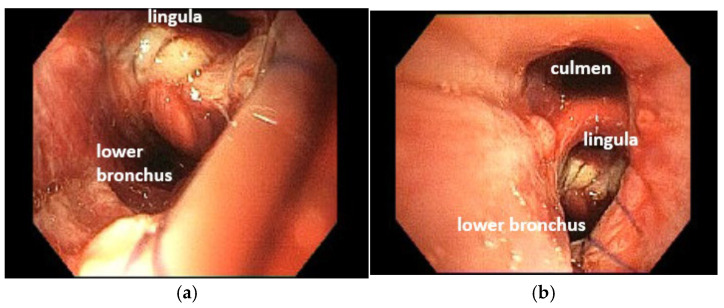
Bronchoscopic evaluation at 7 days post-surgery (**a**,**b**).

**Table 1 cancers-17-02156-t001:** Preoperative data.

Variables		Values N = 25
Sex	Male	10 (40%)
	Female	15 (60%)
Age, y	Median	50
	Minimum, maximum	26–77
Comorbidities	Arterial hypertension	5 (20%)
	COPD	4 (16%)
	Obesity	2 (8%)
	Heart diseases	3 (12%)
Clinical presentation	Cough	10 (40%)
	Recurrent respiratory infections	8 (32%)
	Dyspnoea	6 (24%)
	Chest pain	3 (12%)
	Haemoptysis	3 (12%)
	More than 1 symptom	10 (40%)
	Asymptomatic	5 (20%)
Laser treatment		7 (28%)

**Table 2 cancers-17-02156-t002:** Operative results.

Variables	Results n (%) or n
Median length of resected bronchus (cm)	2.1 (range 1.2–2.9)
Median operative time (min)	220 (range 120–330)
OR extubation (pts)	25 (100%)

OR, operating room; pts, patients.

**Table 3 cancers-17-02156-t003:** Postoperative results.

Histology	Number (%)	pN		R1	Disease Relapse	Right Main Bronchus	Left Main Bronchus	Intermediate Bronchus	Neo-Interlobar Carina
TC	20	20	N0	3	1	5	12	3	5
		1	N1						
AC	3	2	N0	-	1	1	-	2	-
Mucoepidermoid carcinoma	2	2	N0	-	1	-	2	-	-
Total	25	24	N0	3	3	6	14	5	5
		1	N1						

**Table 4 cancers-17-02156-t004:** Perioperative results and complications.

ICU monitoring (pts)	8 (32%)
Prolonged intubation	-
Tracheotomy	-
Median draining time (days)	5 (range 1–12)
Median hospital stays (days)	8 (range 3–13)
Postoperative morbidity	4 (16%)
Atelectasis	1
Atrial fibrillation	1
Hemothorax	1
Anemization	1

ICU, intensive care unit.

## Data Availability

All relevant data are within the manuscript.

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
