# Peer review of "Parenchyma-Sparing Bronchial Sleeve Resection in Low-Grade Malignant Diseases"

_cancers, 2025, doi:10.3390/cancers17132156_

Round 1
Reviewer 1 Report
Comments and Suggestions for Authors
The authors present results of an analysis, describing their experiences in resections of bronchial carcinoids and mucoepidermoid carcinoma, whithout resections lung parenchyma.
They spend a lot of effort in preparation of this manuscript and are according to the STROBE checklist.
The manuscript is very well written and the explanations are comprehensible. Grammar and syntax are fine.
I got few annotations:
- The authors treated 25 patients with this rare procedure, which is quite a big group.
- Some results are markable. 12% of the patients underwent R1 resection, which should not be the aim of resection. Luckily there were no local recurrences. The duration of chest tube is longer than in most manuscripts, but these results are realistic.
- Discussion is way to long. The authors should focus on the most important results ans discuss these. In actual version, they discuss several topics, but each topic is treated superficial.
- The cases are presented in a good way and the pictures are clear.
Reviewer 2 Report
Comments and Suggestions for Authors
Dear Editor and Authors,
As a practicing thoracic surgeon I was with quite an interest that I read and evaluated the manuscript titled "Parenchyma sparing bronchial sleeve resection in low-grade malignant diseases" by Dr. Salimbene and her colleagues.
In this work the authors present their experience in performing sleeve/bronchoplasty resection for treating low grade endobronchial carcinoma which most of the time and in this case represent mainly carcinoid!
The concept the authors propose is not actually so novel as a number of thoracic surgeons (this reviewe included) have performed sleeve resections or bronchotomies to excise low grade endobronchial tumours with excellent results. This has been reported in the literature as well so I would suggest that a bit of toning down on the "pushing" on the novelty and uniqueness of the technique!!
Overall the methodology and setup of the study seems appropriate and correct. This a single institution cohort analysis.
I have the following comments:
- The abstract needs re-working as there are a number of superflous information that don't need presenting at that stage. For example sutture technique and type of procedure.
- The number of patients reported is actually small at 25 cases. Of course the pathologhies that would be ameable to such a technique are relatively uncommon so this could be a mitigating factor.
- How where the data collected for the analysis? The authors need to address this issue. Where data collected from a research database or via chart review?
- Section 2.2 is more appropriate for the results section (including the table).
- Did the authors utilize three-dimensional reconstruction for surgical planing as they suggest and in how many cases?
- Why was a sternotomy performed in one case?
- What about shortening of the bronchus and end mismatch! How did the authors deal with these technical issues?
- What kind of vital flap was used to protect the bronchus? Did the authors create a intercoastal pedicle flap?
- Line 118, what does "to release the bronchial sutture mean"?
- How many stations where sampled in the authors systematic lymphadenectomy?
- Since VATS, VA ECMO and HFJV where not used in the patients they do not need to be mentioned in the table 2!
- Line 136 - ischemic suffering areas needs some re-wording.
- Some of the elements in the material and methods section are better to be included in the results section. For example laser treatment of patients, operative data - table 2,
- The discussion is quite good but I fail to see the point of including Case 1 and 2 other that presenting some nice operative photos!!
In conclusion, this is an interesting and well written up/presented study. I have a few minor comments and clariffications to imrove the work.
Kind regards,
Reviewer 3 Report
Comments and Suggestions for Authors
The authors reported their work named "Parenchyma Sparing Bronchial Sleeve Resection in Low-Grade Malignant Diseases". This retrospective multicenter study evaluates the safety and efficacy of lung-sparing bronchoplastic procedures for low-grade endobronchial neoplasms in 25 patients. Key findings include a 100% 5-year overall survival (OS) and 80% disease-free survival (DFS), with no local recurrences and low postoperative morbidity. While the results are promising, the study’s retrospective design, small sample size, and methodological inconsistencies warrant careful interpretation.
Strengths
- Clinical Relevance: Addresses a gap in surgical options for low-grade endobronchial tumors, advocating for parenchyma preservation.
- Technical Detail: Comprehensive description of surgical techniques, anastomotic methods, and perioperative management.
- Outcome Data: Favorable survival rates and low complication rates (e.g., no bronchopleural fistulas or stenosis) align with existing literature.
- Multidisciplinary Approach: Integration of preoperative imaging, intraoperative bronchoscopy, and postoperative surveillance protocols.
Weaknesses and Concerns
- Sample Size: A small cohort (n=25) limits statistical power, particularly for subgroup analyses (e.g., R1 resections, histologic subtypes).
- Follow-Up Duration: Median follow-up of 37 months (range: 1–88 months) challenges the validity of the reported 5-year survival rates. Kaplan-Meier estimates may overextrapolate incomplete data.
- Lack of Comparative Data: Absence of a control group (e.g., lobectomy patients) weakens assertions about superiority or equivalence of parenchyma-sparing techniques.
- R1 Resections: Three patients had positive margins, yet no local recurrences occurred. The "watch and wait" strategy requires further justification, as adjuvant therapy guidelines for such cases remain unclear.
- Retrospective Design: Potential for selection bias and unmeasured confounders (e.g., surgeon expertise, patient selection criteria).
Methodological Issues
- Statistical Analysis: While appropriate tests were used, the small cohort may underpower analyses (e.g., non-significant DFS differences between margin lengths).
- Ethical Oversight: The statement "This article does not contain any study with human participants" conflicts with the description of patient data collection. Institutional review board approval details should be explicitly stated.
- Inconsistencies:
- Typographical errors (e.g., "canon de fusil" instead of "canon de fusée").
- Reference formatting inconsistencies (e.g., mixed citation styles, missing journal italics).
Recommendations
- Expand Limitations Section: Discuss the impact of the small sample size, retrospective design, and potential biases.
- Validate Survival Data: Provide a minimum follow-up duration for inclusion in survival analysis (e.g., ≥5 years for 5-year OS/DFS).
- Comparative Analysis: Include a control group (e.g., lobectomy patients) in future studies to strengthen conclusions.
- Address R1 Management: Justify the "watch and wait" approach with references to existing guidelines or comparable studies.
- Please add patients at risk to your Kaplan Meier curves.
- Please try to merge the figures together and make multi-panels figures.
- Ethical Compliance: Revise the institutional review board statement to confirm approval for human data use.
- There are numerous typographical and grammatical issues throughout the manuscript (e.g., “bronchscopic” instead of “bronchoscopic”; inconsistent use of spacing before punctuation; missing or incorrect article use).
- Improve consistency in terminology (e.g., “term-to-end” should be “end-to-end”; “latero-terminal” is unclear—consider using “end-to-side” if that is the intended meaning).
- Clarify abbreviations at first use (e.g., PDS, TC, AC).
Round 2
Reviewer 2 Report
Comments and Suggestions for Authors
Dear Editor and Authors,
I re-evaluated the revised manuscript titled "Parenchyma sparing bronchial sleeve resection in low-grade malignant diseases".
The authors have addressed some but not all of the issues raised. Please see below:
They have not addressed my comments number 1 and 2 adequately (not at all actually). They have not changed their text or added any new comments about them!!
Comment 3 was addressed adequately.
Comment 4 was addressed but I suggest that the authors add a small prelude in their results section prior to presenting the old section 2.2 now 3.1 directly! They start with the table with no intro like "Patient demographics are presented in Table 1." ect
Comment 5 was addressed but I am still confused! Did they use imaging reconstruction or not in these cases? From their answer I believe they did not!
Comment 6 was addressed in their response but on line 113 they did not add "because of a combined procedure - what procedure??"
Comment 7 was addressed well.
Comment 8 was addressed and possibly was addressed from the begining.
Comment 9 I am not sure how a lymphadenectomy "releases the structures and creates a tension free anastomosis" but anyway!
Comment 10 was addressed.
Comment 11 was addressed.
Comment 12 was addressed.
Comment 13 was addressed.
Comment 14 was not addressed. This is a research (case series) article and NOT a case report so to include case reports within it is not appropriate!! I understand the authors want to show some "Nice Photos" to showcase their work!! I am a surgeon myself but I recomment that they either write a seperate case report OR include SOME of the photos in their description of the technique section!!
In conclusion, the manuscript has been improved but not yet to the level to be acceptable for publication! I suggest a minor revision of the issues raised!
Comments on the Quality of English LanguageLanguage needs some major editing and proofing by a profesional/native language speaker!
